# Single-Step Self-Assembly of Zein–Honey–Chitosan Nanoparticles for Hydrophilic Drug Incorporation by Flash Nanoprecipitation

**DOI:** 10.3390/pharmaceutics14050920

**Published:** 2022-04-22

**Authors:** Jorge Loureiro, Sónia P. Miguel, Inês J. Seabra, Maximiano P. Ribeiro, Paula Coutinho

**Affiliations:** 1CPIRN-IPG—Center of Potential and Innovation of Natural Resources, Polytechnic Institute of Guarda, Av. Dr. Francisco de Sá Carneiro, No. 50, 6300-559 Guarda, Portugal; jcloureiro97@gmail.com (J.L.); spmiguel@ipg.pt (S.P.M.); mribeiro@ipg.pt (M.P.R.); 2CICS-UBI—Health Sciences Research Centre, University of Beira Interior, Av. Infante D. Henrique, 6200-506 Covilhã, Portugal; 3Bioengineering Department, Lehigh University, Bethlehem, PA 18015, USA; ijs218@lehigh.edu

**Keywords:** zein, chitosan, self-assembly, nanoparticles, flash nanoprecipitation

## Abstract

Zein- and chitosan-based nanoparticles have been described as promising carrier systems for food, biomedical and pharmaceutical applications. However, the manufacture of size-controlled zein and chitosan particles is challenging. In this study, an adapted anti-solvent nanoprecipitation method was developed. The effects of the concentration of zein and chitosan and the pH of the collection solution on the properties of the zein–honey–chitosan nanoparticles were investigated. Flash nanoprecipitation was demonstrated as a rapid, scalable, single-step method to achieve the self-assembly of zein–honey–chitosan nanoparticles. The nanoparticles size was tuned by varying certain formulation parameters, including the total concentration and ratio of the polymers. The zein–honey–chitosan nanoparticles’ hydrodynamic diameter was below 200 nm and the particles were stable for 30 days. Vitamin C was used as a hydrophilic model substance and efficiently encapsulated into these nanoparticles. This study opens a promising pathway for one-step producing zein–honey–chitosan nanoparticles by flash nanoprecipitation for hydrophilic compounds’ encapsulation.

## 1. Introduction

Recent years have witnessed a growth in polymer science and nanotechnology [1,2,3,4]. In particular, nanoparticles (NPs) have received a great deal of attention since they are capable of incorporating drugs, allowing to increase drug bioavailability and cellular uptake [5], targeting [6] or even enhancing diagnosis techniques [7]. Among the different types of NPs produced and reported in the literature, polymeric NPs have various advantages since they can either carry drugs encapsulated within the particle, adsorbed on the surface, or even chemically linked to the surface. In addition, polymeric NPs are usually biodegradable [8,9].

Zein is a water-insoluble prolamin obtained from corn that is generally recognized as safe (GRAS) by the Food and Drug Administration (FDA) with a wide range of applications [10,11,12]. This natural amphiphilic polymer with three–fourths lipophilic and one-fourth hydrophilic amino acid residues is usually dissolved in ethanolic solutions (higher than 70% *v*/*v*) or highly alkaline solutions (pH > 11) [13] and may be assembled in different ways and encapsulate either polar or non-polar drugs [14]. These are unique properties used in the development of NPs. Despite these remarkable characteristics, zein NPs tend to aggregate when in aqueous suspension or even after lyophilization, mainly due to their hydrophobic character [10,14].

Nevertheless, this limitation can be overcome by applying a coating stabilizer, such as chitosan, sodium caseinate and polyvinylpyrrolidone [15]. Apart from the possibility of chitosan acting as a stabilizing agent for zein nanoparticles by reducing aggregation and sedimentation through an increase in electrostatic repulsion among particles, this natural and versatile polymer is also recognized as GRAS by the FDA [16,17]. Furthermore, chitosan versatility is assured due to its functional groups, namely hydroxyl and amine groups, which are responsible for its unique physicochemical and biological properties such as its biocompatibility, biodegradability, antimicrobial activity, mucoadhesive and non-immunogenicity [18,19,20].

Considering these intrinsic properties of chitosan, it has already been used to produce NPs incorporating ß-carotene by flash nanoprecipitation (FNP), in which it is absorbed on the particles’ surface, providing steric and electrostatic stabilization [21]. This pioneering work highlighted the possibility to produce NPs without using organic solvents. Hence, rapid (i.e., within milliseconds), single-step, scalable methods to produce NPs stabilized by chitosan would facilitate the broad application of such particles [22].

In addition, the FNP technique is a well-established method for simple, rapid, polymer-directed self-assembly of NPs for efficient encapsulation of functional agents, as described in detail in the review performed by Saad and Prud’homme [23]. FNP presents several advantages such as its high loading efficiency (typically >80%), short production time (milliseconds), narrow size distribution, and easy scale-up [24,25,26].

The main goal of the present work was to demonstrate the effectiveness of FNP to produce zein–honey–chitosan NPs. The best formulation parameters, based on the effect of NPs’ size and ζ-potential, were examined. Additionally, the honey with excellent biological properties (antibacterial, anti-inflammatory and antioxidant [27]) was included in NPs composition. In addition, the coating of NPs with a mucoadhesive polymer will prompt their application as a drug delivery system in mucosa tissues.

## 2. Materials and Methods

### 2.1. Materials

Absolute ethanol and glacial acetic acid were obtained from VWR International (Porto, Portugal). Bovine serum albumin and fetal bovine serum (FBS) were obtained from Biowest (Nuaillé, France). Chitosan (degree of deacetylation of 75–85% and molecular weight of 50–190 kDa), Dulbecco’s modified Eagle’s medium-high glucose (DMEM-high glucose), Dulbecco’s modified Eagle’s medium/nutrient mixture F-12 ham (DMEM-F12), Fluorescein 5-isothiocyanate (FITC), sodium dodecyl sulfate (SDS), sulforhodamine B (SRB) and trypsin were purchased from Sigma-Aldrich (Lisboa, Portugal). Hoechst 33342^®^ was purchased from Invitrogen (Carlsbad, CA, USA). Honey was obtained from local producers of Guarda, Portugal. L-ascorbic acid (Vitamin C) was obtained from AppliChem (GmbH, Darmstadt, Germany). Penicillin–streptomycin–amphotericin B was obtained from Lonza Walkersville, Inc. (Walkersville, MD, USA). Trichloroacetic acid (TCA) was obtained from Prolab (Leuven, Belgium). Trypsin powder, porcine 1:250 was obtained from SAFC (Sintra, Portugal). Zein was obtained from Acros Organics (Geel, Belgium).

### 2.2. Nanoparticles Assembly and Characterization

In this study, NPs were obtained using a custom-made confined impinging jet mixer based on the model described by Han, et al. [28]. Zein (2.5 mg/mL) and chitosan (10 mg/mL) were dissolved in ethanol (80% *v*/*v*) and glacial acetic acid aqueous solutions (1% *v*/*v*), respectively. Honey was either dissolved in the zein solution or water in a concentration of 10 mg/mL. Four experiments were performed in triplicate (Table 1), always using the ethanolic stream containing zein as one of the streams and water or a chitosan solution as the second stream. In the first experiment, the zein solution (2.5 mL) was mixed against an equal volume of water to evaluate the characteristics of empty NPs. Then, honey-loaded NPs were obtained by dissolving it either in water or in the zein solution. Finally, chitosan was introduced to stabilize the honey-loaded zein NPs.

The resulting solution of the two colliding streams was collected in 45 mL of Milli-Q water, achieving a final volume of 50 mL (Figure 1). The addition of honey either to the zein or to water streams and the incorporation of chitosan was performed to evaluate the influence of those factors on the size and ζ-potential values.

The selected NP formulation (zein–honey–chitosan) was collected at different pH (3, 7 and 10), and size distribution and ζ-potential were evaluated monitored for long-term stability. Vitamin C was used as a hydrophilic compound model to be encapsulated in the NP and was dissolved into the zein–honey 80% ethanolic solution.

Size distributions were reported by determining the z-average and mean hydrodynamic size (average of four measurements). The analysis was performed using a ZetaSizer Nano-ZS (Malvern Instruments, Malvern, UK) at 25 °C and a scattering angle of 173°. An aliquot was taken and the size was directly determined by dynamic light scattering. The polydispersity index (PDI), which is a measure of the breadth of size distribution, was determined from the instrument software (appropriate for samples with PDI < 0.4) and was used as a parameter for particle uniformity, where the solvent used was water. NPs stability was evaluated by monitoring the NPs’ size distributions at various times (7, 15, 30, 60 and 90 days) after mixing and stored at 4 °C.

#### 2.2.1. FTIR

FTIR analysis was performed on freeze-dried NPs to characterize its composition. The spectra were acquired using an average of 128 scans, a spectral width ranging from 600 cm^−1^ to 4000 cm^−1^ and a resolution of 32 cm^−1^. All the samples were mounted on a diamond window, and the spectra were recorded using a Nicolet iS10 FTIR spectrophotometer (Thermo Scientific, Waltham, MA, USA). The FTIR spectra of the raw materials used were also acquired for comparative purposes.

#### 2.2.2. Drug Encapsulation Efficiency and Release Profile

The encapsulation and loading efficiency of Vitamin C into nanoparticles was determined through the precipitation of the nanoparticles by using 10 k Amicon^®^ Ultra-2 Centrifugal Filter Devices (Merck Millipore, Darmstadt, Germany). Following the manufacturers’ protocol, it was possible to recover the supernatant containing the free Vitamin C. Then, the Vitamin C was quantified by UV–Vis spectrophotometry Multiskan GO (Thermo Scientific, Waltham, MA, USA) using a standard absorbance curve (y = 10.086x − 0.0005, R^2^ = 0.99).

The UV absorbance spectra of nanoparticles without Vitamin C was also recorded for control purposes. The encapsulation efficiency (*EE*) was determined using the following Equation (1).
(1)$$EE\left(\%\right)=\frac{W_{i}-W_{s}}{W_{i}}$$
where *W_i_* is the total amount of Vitamin C added into zein–honey–chitosan nanoparticles and *W_s_* is the amount of Vitamin C in the supernatant.

#### 2.2.3. Characterization of Vitamin C Release Profile from Nanoparticles

Afterwards, the amount of Vitamin C released from the nanoparticles solution was measured according to a method previously reported in the literature [29]. The nanoparticles were incubated at 37 °C under agitation to mimic the physiological conditions. At different time points, the nanoparticles sample were separated from Vitamin C by using Amicon^®^ Ultra-2 Centrifugal Filter Devices (10 k). Afterwards, the supernatant absorbance was measured at 288 nm using UV/Vis spectrophotometry to determine the amount of Vitamin C released. All experiments were performed in triplicate.

### 2.3. In Vitro Assays

#### 2.3.1. Evaluation of Cytotoxicity

Normal human dermal fibroblasts (NHDFs) purchased from PromoCell (Labclinics, S.A., Barcelona, Spain) and immortalized human cervical cancer cell line HeLa purchased from American type culture collection (ATCC, Middlesex, UK) were used as cell models and maintained in a humidified atmosphere at 37 °C in 5% CO_2_ in DMEM-F12 and DMEM-high glucose, respectively, supplemented with 10% FBS, and 1% penicillin–streptomycin–amphotericin B. To evaluate the cytotoxicity of NPs, NHDF and HeLa cells were seeded in 96-well plates (1 × 10^4^ cells/well). Before each cytotoxicity experiment, the cell adhesion and proliferation were allowed for 24 h. After that, the culture media were replaced with NPs and honey solutions of different concentrations (100–800 µg/mL). SDS was used as a positive control to induce cell death (K+), whereas untreated cells were used as negative control (K-). Cell growth was monitored using an Optika inverted light microscope equipped with an Optikam B5 (Bergamo, Italy) digital camera.

The SRB assay is a colorimetric assay that quantifies cell viability by binding SRB to cell proteins and providing accurate results [30]. Briefly, following a 24 h and 72 h treatment incubation, the medium was aspirated, and cells were fixed with 25 µL of 10% (*w*/*v*) TCA and kept at 4 °C for 1 h. Furthermore, a gentle washing step with milli-Q water and cells were stained for 10 min by adding 50 µL of SRB (0.4% *w*/*v*) in acetic acid (1% *v*/*v*) at room temperature. An unbound SRB stain was removed by washing three times with acetic acid (1% *v*/*v*) and left to dry overnight. SRB dye was then solubilized by adding 100 µL of Tris base (10 mM, pH = 10.5). Its absorbance was measured at 510 nm using a microplate reader (Multiskan GO by Thermo Scientific). Results are reported in average ± SD (*n* = 5).

#### 2.3.2. Nanoparticles Cellular Uptake

The uptake of the NPs by NHDF cells was characterized through confocal laser scanning microscopy (CLSM). To enable the NPs visualization, these were loaded with FITC, as described by Guo, et al. [31]. Briefly, the FITC dye (400 µg/mL) was incorporated into NPs during the flash nanoprecipitation procedure. A step of dialysis was further performed, aiming to remove the possible unloaded FITC dye.

Then, 2 × 10^4^ NHDF cells were seeded on µ-Slide 8 well Ibidi imaging plates (Ibidi GmbH, Gräfelfing, Germany) and incubated for 24 h at 37 °C in a humidified atmosphere with 5% CO_2_. Afterwards, 300 µg/mL of NPs stained with FITC was added to each well and incubated for 2 and 4 h. Subsequently, the seeded cells were washed with PBS, fixed with a 1:1 mixture of methanol/ethanol for 10 min and rinsed with PBS. For cell nucleus staining, Hoechst 33342 was used. The CLSM images were obtained using a Zeiss LSM 710 Confocal microscope (Carl Zeiss AG, Oberkochen, Germany). The image analysis was performed in the Zeiss Zen 2010 software.

Additionally, the uptake of the NPs was also determined by fluorescence spectroscopy according to a method described by Moreira, et al. [32] with slight modifications. Briefly, the cells were seeded into 96-well plates at a density of 1 × 10^4^ cells/well and cultured for 24 h in a humidified atmosphere at 37 °C in 5% CO_2_. After this period, the culture media were removed, and the cells were incubated with FITC-stained zein–honey–chitosan NPs at a concentration of 300 μg/mL for 4 h. Then, the cells were washed with ice-cold phosphate-buffered saline (PBS) and lysed with 1% Triton X-100 in PBS for 30 min at 37 °C. Cells only incubated with medium were used as control. Afterward, FITC fluorescence (λex = 490 nm and λem = 520 nm) was quantified using a Spectramax Gemini XS (Molecular Devices LCC, San José, CA, USA).

### 2.4. Statistical Analysis

Each experiment was set in triplicate and resulted from three independent assays. A two-way ANOVA followed by a Bonferroni and Tukey post hoc test (*p* < 0.5) was used for the cytotoxicity and stability assays, respectively. Data were considered significant at *p* < 0.5 (*). The results were obtained using GraphPad Prism 7 software (GraphPad, San Diego, CA, USA).

## 3. Results and Discussion

### 3.1. Synthesis and Characterization of Zein–Honey–Chitosan Nanoparticles

FNP technology has become increasingly popular in producing drug-loaded NPs because of its high reproducibility and reliability [13,33]. In this work, NPs with a low hydrodynamic diameter (<200 nm) were successfully produced by FNP (Table 2 and Figure 2).

Zein NPs presented a size of 126.20 ± 1.071 nm, a very low PDI (0.007 ± 0.027) and a negative ζ-potential (−12.40 ± 0.526 mV). When honey was added in the opposite aqueous stream, the size and PDI significantly increased (*p* < 0.0001) to 185.20 ± 1.564 nm and 0.107 ± 0.023, respectively. On the other hand, when zein and honey were placed in the same stream, the size variation was not statistically significant compared to the empty NPs (which decreased from 126.20 ± 1.071 nm to 125.60 ± 0.885 nm). However, concerning PDI results, an increase from 0.007 ± 0.027 to 0.086 ± 0.014 (*p* = 0.0004) was verified; nevertheless, maintaining a PDI lower than 0.4. Finally, to stabilize the NPs, chitosan was added as a coating agent. The NPs with zein and honey in one stream and chitosan in the other stream exhibited a size of 148.1 ± 3.444 nm, a PDI of 0.300 ± 0.028 and a positive ζ-potential (48.30 ± 1.150 mV).

These values showed a significant increase in the size and PDI of the zein–honey–chitosan NPs when zein and honey were placed on the same steam. Regarding the ζ-potential, there were significant differences when comparing empty NPs and those produced with zein in steam one and honey in steam two. When honey and zein were in the same steam, there were no statistically significant differences in the empty NPs. For ζ-potential, it was observed that values variated significantly from slightly negative (−12.40 ± 0.526 mV) to a more positively charged (48.30 ± 1.150 mV) NPs, suggesting that chitosan coated the NPs.

Therefore, the NPs produced in the present study show favorable properties as nano delivery systems, since the nanoparticles: (i) had a size smaller than 200 nm; (ii) a PDI lower than 0.4, which indicates a homogeneous size distribution of NPs; and (iii) a positive ζ-Potential which is ideal for interaction with the negatively charged surface of cells, Shi [34], as well as with the net-negative surface charge of mucin glycoproteins [35]. Moreover, such physicochemical properties are described as promising carriers for different drugs associated with longer blood circulation times and avoiding accumulation into organs such as the spleen, liver and lungs [36,37,38,39]. Apart from the blood administration, such NPs can also act as a mucosal drug delivery system which avoids the first-pass hepatic metabolism and the degradation by gastrointestinal enzymes [40].

Considering the obtained results (size under 200 nm, PDI of 0.3 and positive ζ-potential), the formulation with zein, honey and chitosan (zein–honey–chitosan NPs) was chosen for further assays, namely stability evaluation, FTIR spectra analysis and in vitro assays [22].

#### 3.1.1. FTIR

After the NPs production, FTIR analysis was performed to confirm that FNP maintains the structural integrity of the raw materials (chitosan, zein and honey). The FTIR spectra of raw materials were acquired and compared with those of the NPs, as shown in Figure 2.

The zein FTIR spectrum exhibited its typical characteristic peak at 3292 cm^−1^, assigned to O–H stretching vibrations, and peaks at 1643 and 1521 cm^−1^, corresponding to C=O stretching and N–H bending vibrations, respectively [41]. The chitosan spectrum showed major bands at 3351 cm^−1^ (O–H and N–H stretching vibrations); at 2872 cm^−1^ (C–H stretching vibrations); at 1649 cm^−1^ (C=O stretching); 1589 cm^−1^ (N–H bending vibration); 1375 cm^−1^ (C–O stretching vibration of the primary alcohol group); and 1026 cm^−1^ (C–O–C glycosidic bond) [42]. In turn, the FTIR spectrum of honey showed the characteristic peaks at 3672 cm^−1^ (O–H stretching vibrations); 2977 cm^−1^ (C–H stretching vibrations); 1771 cm^−1^ (C=O stretching); and 1063 cm^−1^ (C–O stretching) [43]. By analyzing the FTIR spectrum acquired for zein–honey–chitosan NPs, it is possible to verify that the characteristic peaks of the involved raw materials are also present, indicating that the FNP did not affect their chemical structure.

#### 3.1.2. Evaluation of the Stability of Zein–Honey–Chitosan NPs

NPs’ stability is a critical parameter for translation into clinical practice. Stability studies provide information about the preservation of NPs’ properties (size, PDI and ζ-potential) under controlled conditions.

As such, the NPs were stored at 4 °C for 90 days, and their stability was monitored by measuring the size, PDI and ζ-potential at specific time points. The effect of the pH of the collecting solution was investigated by collecting the NPs in aqueous solutions with three different pH values (3, 7 and 10).

Figure 3 shows that the NPs collected in the solution whose pH = 3 exhibited an initial mean diameter of 117.0 ± 0.332 nm, which remained stable for 90 days. The NPs collected at pH 7 and pH 10 displayed an initial hydrodynamic diameter of 133.9 ± 0.808 nm and 116.1 ± 1.89 nm, respectively, which significantly increased after 60 days of incubation (*p* < 0.05). The mean sizes of these NPs became larger after 2 months of storage, which can be attributed to particle aggregation. Such facts could have resulted from reduced electrostatic repulsion among the NPs, decreasing the positive surface charge [44]. No statistically significant differences were found for NPs at day 0, collected at solutions with different pH, and for all pH evaluated NPs were stable for at least 30 days. This can be related with the low swelling rate and be considered as an advantage to obtain a stable carrier system able to incorporate and deliver drugs at a controllable rate.

On the other hand, when evaluating stability for 90 days, NPs collected in pH = 3 were stable for more time (90 days) than those collected in pH = 7 and pH = 10 (30 days). These differences in stability may be explained by the relation of pH values and the presence of H^+^/OH^−^, in which if the pH is acid, there is more H^+^ and if the pH is alkali, then there are more OH^−^. In pH = 3, there is a high presence of H^+^ ion, which will reduce the NH_2_ groups of chitosan to NH_3_^+^, and consequently increase the electrostatic repulsion between NPs as there are more NH_3_^+^ than NH_2_. On the other hand, if the pH increases, the H^+^ ion concentration will decrease. Consequently, fewer NH_2_ groups will be reduced to NH_3_^+^, reducing the electrostatic repulsion, which translates into reduced stability for the NPs collected in pH = 7 and pH = 10 [45].

Regarding PDI, the values for the different pH were not statistically significant at day 0, indicating that the technique produced very similar particles regardless of the pH value of the collecting solution. When comparing 0 vs. 7, 7 vs. 15, 15 vs. 30, 30 vs. 60 and 60 vs. 90, NPs collected at pH = 7 and pH = 10 did not have statistically significant variations until 30 vs. 60 days. However, a statistically significant variation for pH = 3 was registered at 0 vs. 7 days.

Concerning the ζ-potential values, these showed a statistically significant tendency to decrease when comparing 0 vs. 7, 7 vs. 15, 15 vs. 30, 30 vs. 60 and 60 vs. 90 days. For pH = 3, the values significantly decreased from 50.0 ± 0.7 to 35.2 ± 0.6, except for day 15 vs. day 30. The decrease was statistically significant for the other pH values in every time comparison. This may indicate that the nanoparticles are aggregating.

Considering all results, the collecting solution at pH = 3 presented the most promising results for the stability of NPs. Such an occurrence can be attributed to its acidic environment, filled with H+ ions, which increases the electrostatic interactions with the NH_3_^+^ chitosan groups, and consequently increases repulsion [46].

### 3.2. Vitamin C Release Profile

Herein, Vitamin C (a model hydrophilic compound) was selected to be loaded into the nanoparticles aiming to evaluate their potential as a hydrophilic drug delivery system.

The EE obtained for the Vitamin C incorporation into the nanoparticles was 82.4 ± 0.12%, indicating that hydrophilic compounds can be encapsulated into zein–honey–chitosan nanoparticles using the flash nanoprecipitation technique.

Afterwards, the in vitro release of Vitamin C from NPs was investigated in a simulated physiological environment (pH = 7) following the indications described in the literature, where this pH value is considered the most adequate to predict the performance of systems [47,48,49,50,51,52].

As shown in Figure 4, the Vitamin C release profile exhibits a controlled and sustained release profile during at least 24 h. Such a release profile can be related to the surface-to-volume ratio of the nanoparticles that favor water adsorption and hence the diffusion of Vitamin C from the nanocarriers. Moreover, the presence of chitosan in the nanocomposite particle carrier results in mucoadhesion and promotes the bioavailability of the biomolecule by interacting with the mucosa [53]. In addition, the controlled release behavior of Vitamin C from nanoparticles can also be correlated to their low swelling rate, which is responsible for controlling the diffusion of drugs when the water invades the polymers chains [54].

### 3.3. In Vitro Assays

The SRB assay was used to investigate the cytotoxicity of NPs produced to determine the protein content on NHDF and Hela cells after incubating different concentrations of NPs [30]. In addition, honey cytocompatibility was also evaluated in contact with both types of cells.

The SRB assay showed (Figure 5) that the honey at concentrations lower than 600 µg/mL did not compromise the NHDF viability, whereas at the same concentration, it promoted a slight decrease in the cell viability of HeLa cells. Such results indicate that honey can exhibit anticancer activity, as previously described by Fauzi et al. and Salleh et al. [55,56]. After 72 h of incubation, the NPs demonstrated a highly compatible character for NHDF in comparison with the honey. In turn, for the HeLa cell line, the NPs presented a preferential accumulation which was verified by toxic effect on the cancer cell line.

On the other side, the NPs showed a biocompatible character for NHDF cells at concentrations lower than 300 µg/mL with a similar effect on HeLa cells.

The microscope images (Figure 6) showed the effect of honey and NPs concentrations on the morphology and proliferation of NHDF and HeLa. These images indicated that cells present the characteristic morphology (at lower concentrations) similarly to the live cells (cells incubated just culture medium). As expected, the biocompatible profile of NPs is related to the natural, biocompatible biopolymers (chitosan, zein) used in NPs production [57].

Together, the results show evidence that the developed NPs are a promising drug delivery system to incorporate hydrophilic drugs for biomedical applications.

After assessing the biocompatibility of zein–honey–chitosan NPs, the NPs cellular uptake by NHDF cells was evaluated by CLSM and fluorescence spectroscopy (Figure 7). The zein–honey–chitosan tracking was achieved by FITC staining. In Figure 7A, it is possible to observe the presence of zein–honey–chitosan NPs (identified with the white arrows) interacting with NHDF cells. Furthermore, the FITC-nanoparticles and non-stained nanoparticles were incubated for 4 h with NHDF cells and then the FITC fluorescence was measured (Figure 7B). The fluorescence spectroscopy studies revealed the internalization of zein–honey–chitosan NPs, which is in agreement with the confocal images.

The cells membrane is the last barrier that the nanoparticles have to overcome to deliver their payload to the cells. The cell internalization studies indicate that the positive surface charges of the particles can favor the nanoparticle interaction with the cellular membrane enhancing the cellular uptake [58]. Such behavior will allow drug-delivery into the cell, thus avoiding the drug premature degradation in the extracellular medium.

## 4. Conclusions

In this study, the production of zein–honey–chitosan NPs by a one single-step and continuous method via FNP was demonstrated. Such a simple, rapid, and versatile method enables the production of NPs with reproducible properties. The polymers ratio was optimized, and the produced NPs displayed a suitable size, PDI and surface charge for biomedical applications. In addition, honey was successfully integrated into NPs, revealing that the collecting solution at pH = 3 presented the most promising potential for assuring the stability of the physical features of NPs.

Furthermore, the NPs exhibited a biocompatible profile when incubated with NHDF cells at concentrations lower than 300 µg/mL. In turn, a slight decrease in HeLa cells’ viability was verified for NPs’ concentration at 600 µg/mL. In addition, the fluorescence images evidenced that the NPs were able to interact with cellular membrane. Such results indicated that the NPs could be exploited for different biomedical applications such as tissue engineering and/or cancer therapy, which can be administrated via mucosa tissue.

Additionally, the nanoparticles’ ability to act as a drug delivery system was evidenced in the drug release studies by incorporating Vitamin C as a hydrophilic compound model.

In short, the present works demonstrated the potential of the FNP method for producing polymeric NPs with physical features appropriated for biomedical applications. In the near future, additional assays (to evaluate the cell cycle profile and cell death mechanisms) will be performed to investigate the anticancer activity of NPs and explore other promising properties to act as a mucosal drug delivery system.

## Figures and Tables

**Figure 1 pharmaceutics-14-00920-f001:**
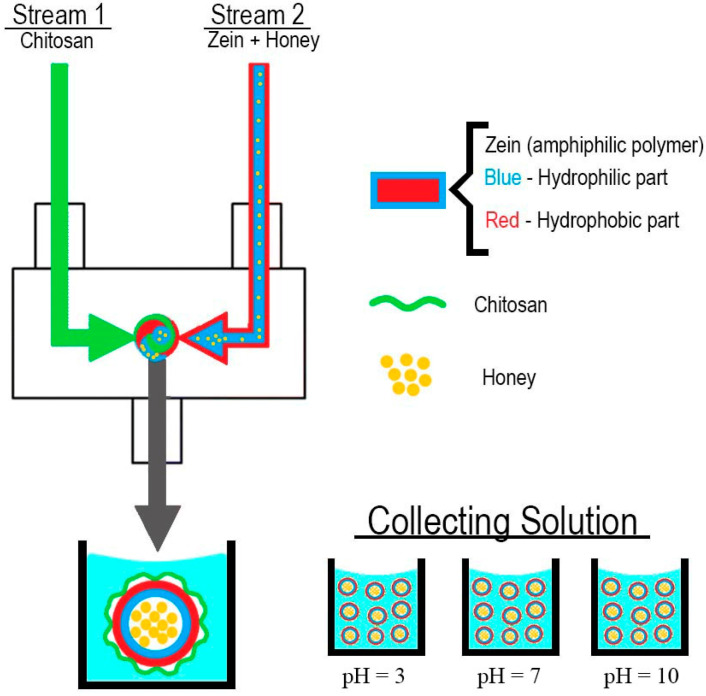
The proposed approach for zein–honey–chitosan NPs synthesis by FNP.

**Figure 2 pharmaceutics-14-00920-f002:**
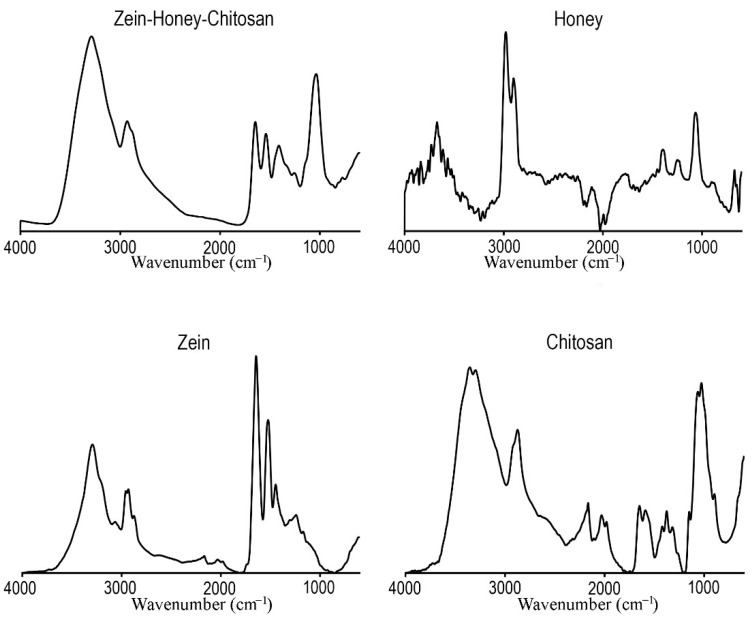
FTIR spectra (transmittance vs. wavenumber) of NPs and individual components used for NPs production.

**Figure 3 pharmaceutics-14-00920-f003:**
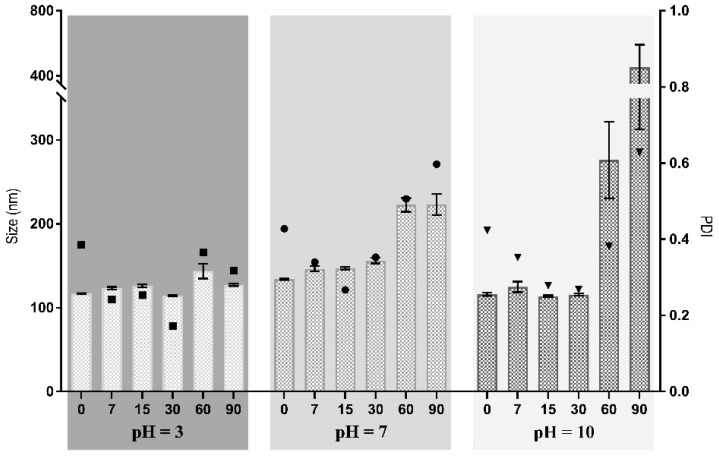
Stability of zein–honey–chitosan NPs in three different collecting solutions for 90 days. White bars and black squares represent the size and PDI of NPs collected in pH = 3, respectively. Light grey bars and black circles represent the size and PDI of NPs collected in pH = 7, respectively. Dark grey bars and black triangles represent the size and PDI of NPs collected in pH = 10. Data are presented as mean ± SEM (*n* = 3).

**Figure 4 pharmaceutics-14-00920-f004:**
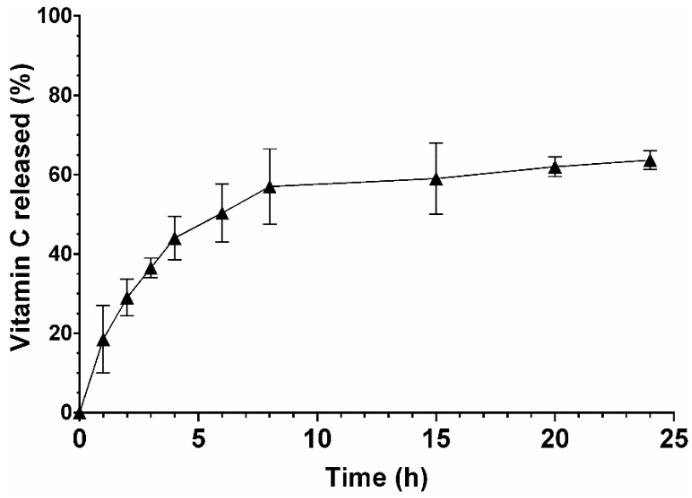
Characterization of the release profile of Vitamin C from nanoparticles at pH 7. Black triangle represents the percentage of Vitamin C released at different timepoints. Data are presented as mean ± SEM (*n* = 3).

**Figure 5 pharmaceutics-14-00920-f005:**
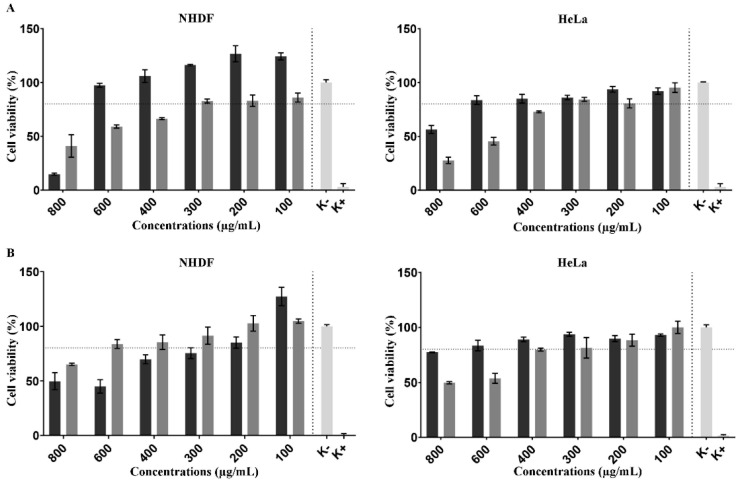
Cell viability determined by the SRB assay of different concentrations of NPs and honey when incubated with HDF and HeLa cells. Data are presented as the mean ± standard deviation (*n* = 5). (**A**) and (**B**) represent the results obtained for 24 h and 72 h, respectively. The black and grey bars correspond to the honey and nanoparticles, respectively. Data are presented as mean ± SEM (*n* = 3).

**Figure 6 pharmaceutics-14-00920-f006:**
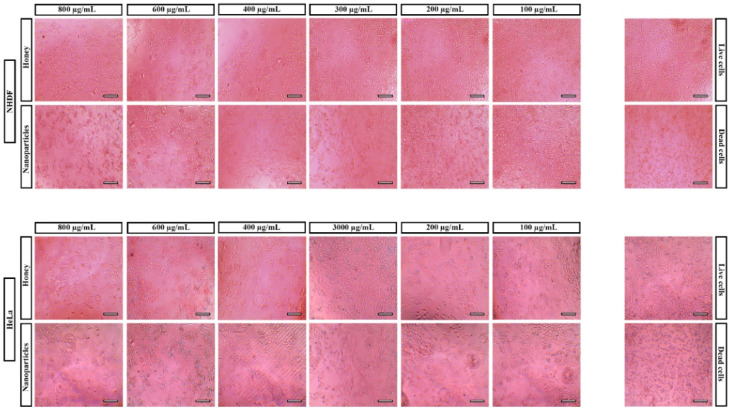
Microscopic images of NHDF and HeLa cells in contact with different concentrations of NPs and honey. Scale bar: 100 µm.

**Figure 7 pharmaceutics-14-00920-f007:**
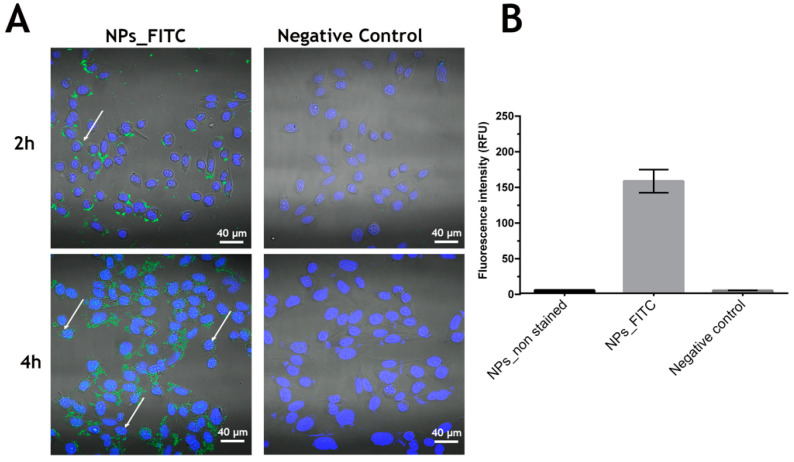
Analysis of NPs uptake in NHDF cells after 2 h and 4 h of incubation and a negative control (**A**) the white arrows point towards the internalized nanoparticles. Blue channel: the Hoechst 33342s stained cell nucleus; green channel: nanoparticles stained with FITC. (**B**) Quantification of the fluorescence intensity of FITC-stained nanoparticles after 4 h of incubation. Scale bar corresponds to 40 µm. Data are presented as mean ± SEM (*n* = 3).

**Table 1 pharmaceutics-14-00920-t001:** Experiments performed in this work with different alternatives in solutions used in stream 1 and stream 2.

Experiment	Stream 1	Stream 2
1	Zein	Water
2	Zein	Water + honey
3	Zein + honey	Water
4	Zein + honey	Chitosan

**Table 2 pharmaceutics-14-00920-t002:** Size, PDI and ζ-potential of the NPs. Data are presented as mean ± SEM (*n* = 3).

Stream 1	Stream 2	Size (nm)	PDI	ζ-Potential (mV)
Zein	Water	126.20 ± 1.071	0.007 ± 0.027	−12.40 ± 0.526
Zein	Water + honey	185.20 ± 1.564	0.107 ± 0.023	−10.50 ± 0.537
Zein + honey	Water	125.60 ± 0.885	0.086 ± 0.014	−12.80 ± 0.351
Zein + honey	Chitosan	148.10 ± 3.444	0.300 ± 0.028	48.30 ± 1.150

## Data Availability

No new data were created or analyzed in this study. Data sharing is not applicable to this article.

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
