# Peer review of "Single-Step Self-Assembly of Zein–Honey–Chitosan Nanoparticles for Hydrophilic Drug Incorporation by Flash Nanoprecipitation"

_pharmaceutics, 2022, doi:10.3390/pharmaceutics14050920_

Round 1
Reviewer 1 Report
This manuscript reported a Zein-honey-chitosan nanoparticle for hydrophilic drug delivery. However, the authors didn't show any data about encapsulation or release of drugs. I recommend the authors to give related data to further demonstrate this nanosystem can effectively load and delivery drugs. Thus, major revision is required.
Author Response
The authors acknowledge the reviewer for pointing this issue. As suggested, the vitamin C was incorporated into nanoparticles, as hydrophilic model. The encapsulation efficiency as well as their release profile along 24 hours were assessed. The data were added to the revised version of manuscript (please see page 9, lines 336- 357 and figure 4).
Reviewer 2 Report
In this manuscript, the authors have studied the development of zein-honey-chitosan nanoparticles (NPs) for drug delivery systems. This article describes how to prepare-/develop a zein-honey-chitosan NPs by using the flash nanoprecipitation strategy. In my view, the manuscript (ID: pharmaceutics-1624600) could be accepted after considering the following comments:
1. The authors should show the loading capacity (encapsulation efficiency) of the nanoparticles.
2. The authors should study-/determine the release profiles (pH-responsive drug release) of different formulations at different pH values.
3. The swelling rate of the NPs should be studied.
4. The cell viability of nanocarriers should be studied after a longer incubation period, i.e. 72 h, in order to investigate an accurate-/real cytotoxicity.
5. The cellular internalization of nanocarriers should be proved and illustrated in order to clarify, i.e., nanoparticle–cell interactions and etc.
6. Fig.5: It is not clear what is happening, i.e., to the cell adhesion, shape, and morphology, etc. The authors should provide a live/dead assay (e.g., Calcein-AM-/PI), and other staining protocols (e.g, actin, nucleus, ...) to investigate cell responses.
Author Response
In this manuscript, the authors have studied the development of zein-honey-chitosan nanoparticles (NPs) for drug delivery systems. This article describes how to prepare-/develop a zein-honey-chitosan NPs by using the flash nanoprecipitation strategy. In my view, the manuscript (ID: pharmaceutics-1624600) could be accepted after considering the following comments:
1. The authors should show the loading capacity (encapsulation efficiency) of the nanoparticles.
2. The authors should study-/determine the release profiles (pH-responsive drug release) of different formulations at different pH values.
Thank you. As previously described, the data for the encapsulation efficiency and release profile of the vitamin C into nanoparticles was included in the revised version of the manuscript (please see page 9, lines 336- 357 and figure 4).
The main goal was the validation of the flash nanoprecipitation technique for the polymeric nanoparticles production with ability to encapsulate hydrophilic compounds (vitamin C was used as model). As suggested, the release profile was assessed at pH 7.4. The majority of the drug release studies were performed at physiological and osmolality conditions (pH 7.4), which is considered the most adequate to predict the performance of systems (Ak, 2021; Bashir, Teo, Ramesh, Ramesh, & Mushtaq, 2018; Leung et al., 2020; Pimenta, Serro, Colaço, & Chauhan, 2019; Tamahkar, Özkahraman, SüloÄŸlu, Ä°dil, & Perçin, 2020; Vigata, Meinert, Hutmacher, & Bock, 2020). In future, depending on the final application of the nanosystem and drug selected, the release studies in different pH values should be considered. Such was clarified in the revised version of the manuscript (please see page 9, lines 342-347)
3. The swelling rate of the NPs should be studied.
The swelling rate of a material is indicative of their ability to absorb fluids, however such property influences the increase of hydrodynamic diameter of nanoparticles during the stability assays. Such can occur due to the interaction of polymers with water molecules through hydrogen bonds. In this work, the stability assays of the zein-chitosan-honey nanoparticles revealed that the hydrodynamic diameter remain stable at least 30 days at different pH evaluated. So, this can be related with the low swelling rate and be considered as an advantage to obtain a stable carrier system able to incorporate and deliver drugs at controllable rate. This was clarified in the revised version manuscript (please see page 8, lines 304-307).
4. The cell viability of nanocarriers should be studied after a longer incubation period, i.e. 72 h, in order to investigate an accurate-/real cytotoxicity.
Thank you for your suggestion. The cell viability results of nanoparticles when incubated with fibroblasts and HeLa cells during 72 hours were added to the revised version of manuscript (please see figure 6).
5. The cellular internalization of nanocarriers should be proved and illustrated in order to clarify, i.e., nanoparticle–cell interactions and etc.
6. Fig.5: It is not clear what is happening, i.e., to the cell adhesion, shape, and morphology, etc. The authors should provide a live/dead assay (e.g., Calcein-AM-/PI), and other staining protocols (e.g, actin, nucleus, ...) to investigate cell responses.
As suggested, the cell internalization of the nanocarriers was characterized through Confocal Laser Scanning Microscopy, where it was possible to verify that the nanocarriers were able to interact with the cell membranes, and the cell shape and morphology was not affected. Moreover, the quantification of the internalized nanoparticles into cells was also determined by measurement of fluorescence intensity of FITC- zein-chitosan-honey nanoparticles and the results corroborate the fluorescence images supporting the nanoparticles internalization. Such results were added to the revised version of the manuscript, please see Figure 7, page 11, lines 397-410.
References
Ak, G. (2021). Covalently coupling doxorubicin to polymeric nanoparticles as potential inhaler therapy: in vitro studies. Pharmaceutical Development and Technology, 26(8), 890-898.
Bashir, S., Teo, Y. Y., Ramesh, S., Ramesh, K., & Mushtaq, M. W. (2018). Rheological behavior of biodegradable N-succinyl chitosan-g-poly (acrylic acid) hydrogels and their applications as drug carrier and in vitro theophylline release. International journal of biological macromolecules, 117, 454-466.
Leung, B., Dharmaratne, P., Yan, W., Chan, B. C., Lau, C. B., Fung, K.-P., . . . Leung, S. S. (2020). Development of thermosensitive hydrogel containing methylene blue for topical antimicrobial photodynamic therapy. Journal of Photochemistry and Photobiology B: Biology, 203, 111776.
Pimenta, A. F., Serro, A. P., Colaço, R., & Chauhan, A. (2019). Optimization of intraocular lens hydrogels for dual drug release: Experimentation and modelling. European Journal of Pharmaceutics and Biopharmaceutics, 141, 51-57.
Tamahkar, E., Özkahraman, B., SüloÄŸlu, A. K., Ä°dil, N., & Perçin, I. (2020). A novel multilayer hydrogel wound dressing for antibiotic release. Journal of Drug Delivery Science and Technology, 58, 101536.
Vigata, M., Meinert, C., Hutmacher, D. W., & Bock, N. (2020). Hydrogels as drug delivery systems: A review of current characterization and evaluation techniques. Pharmaceutics, 12(12), 1188.
Reviewer 3 Report
The paper is poor and in the present form is rejected.
Following some suggested experiments to improve the quality and data consistence of work.
- The authors should perform physical chemical characterizations of their nanoformulations such as TEM, SEM, AFM, etc.
- The authors should perform cellular experiments as uptake, intracellular localization, cell cycle after treatment of nanoparticles, apoptosis, etc.
Author Response
The paper is poor and in the present form is rejected. Following some suggested experiments to improve the quality and data consistence of work.
- The authors should perform physical chemical characterizations of their nanoformulations such as TEM, SEM, AFM, etc.
- The authors should perform cellular experiments as uptake, intracellular localization, cell cycle after treatment of nanoparticles, apoptosis, etc.
The authors thank the reviewer for the important comment. Indeed, TEM analysis are usually performed to characterize the morphology of the nanoparticles. However, the DLS offers a quantitative analysis and is the technique most used for the characterization of the nanoparticles in biomedical applications including in the industry (da Rosa et al., 2015; de Vlieger et al., 2019; Vahedikia et al., 2019).
On the other hand, the SEM and AFM will provide information about the surface morphology of nanoparticles which is important to characterize their interaction with cells. In this field, the obtained confocal microscope images already demonstrate the ability of nanoparticles to interact with the cell membrane compounds. Such phenomenon may be related to the positive surface charge of nanoparticles that will interact with negative charged glycosaminoglycans available on surface membrane through electrostatic interactions. The cell internalization of the nanocarriers was also characterized through Confocal Laser Scanning Microscopy and the fluorescence relative to FITC- zein-chitosan-honey nanoparticles internalized into cells was quantified (please see Figure 7, page 11, lines 397-410).
References
da Rosa, C. G., Maciel, M. V. d. O. B., de Carvalho, S. M., de Melo, A. P. Z., Jummes, B., da Silva, T., . . . Barreto, P. L. M. (2015). Characterization and evaluation of physicochemical and antimicrobial properties of zein nanoparticles loaded with phenolics monoterpenes. Colloids and Surfaces A: Physicochemical and Engineering Aspects, 481, 337-344.
de Vlieger, J. S., Crommelin, D. J., Tyner, K., Drummond, D. C., Jiang, W., McNeil, S. E., . . . Shah, V. P. (2019). Report of the AAPS guidance forum on the FDA draft guidance for industry:“drug products, including biological products, that contain nanomaterials”: Springer.
Vahedikia, N., Garavand, F., Tajeddin, B., Cacciotti, I., Jafari, S. M., Omidi, T., & Zahedi, Z. (2019). Biodegradable zein film composites reinforced with chitosan nanoparticles and cinnamon essential oil: Physical, mechanical, structural and antimicrobial attributes. Colloids and Surfaces B: Biointerfaces, 177, 25-32.
Round 2
Reviewer 1 Report
The manuscript can be accepted at its present form.
Author Response
Thank you!
Reviewer 2 Report
Accept
Author Response
Thank you!
Reviewer 3 Report
Thanks for kindly reply to authors. Morphological characterizations of nanoparticles (TEM, SEM, AFM) are essential analysis for know the interaction of NPs with cells and other biological effects as uptake, intracellular localizations, apoptosis, etc.
DLS give an indication of hydrodynamic size, PdI and zeta potential of NPs. Confocal laser scanning microscopy (CLSM) give information regarding the uptake and intracellular localization. DLS and CLSM not give information’s regarding the morphology, roughness, etc. that are possible to valuate only with other techniques, as TEM, SEM and AFM.
What is the shape and the roughness of Zein-honey-chitosan NPs? In addition, what’s the cell cycle profile after treatment of nanoparticles? There is apoptosis? Please add this information.
Author Response
The authors appreciate your important comments. We agree that the TEM, SEM and AFM are powerful techniques that allows to characterise the surface properties of the materials, as well as DLS. In fact, the DLS is the most used tool for the size characterization of spherical particles (Arenas-Guerrero et al., 2018), and it has been shown that the contribution of rotational diffusion can have important effect on DLS determinations in non-spherical particles, which can result in non-validation of the results as well as higher PDI values (Liu & Xiao, 2012). In this sense, our results, namely the NP hydrodynamic diameter and PDI, validated by DLS technique indicate that the nanoparticles present spherical shape. As for, different chitosan-zein nanoparticles has also been reported in the literature as presenting a spherical shape (Luo, Zhang, Cheng, & Wang, 2010; Luo, Zhang, Whent, Yu, & Wang, 2011).
On the other hand, the surface roughness and materials charge are considered crucial parameters to predict the cell interaction. In this case, the cell interaction is guarantee by the surface charge of the nanoparticles, and the cell internalization studies also confirmed the interaction of the nanoparticles with cell membranes. Moreover, the cell cycle profile and apoptosis mechanisms after the nanoparticles treatment are important to complement the metabolic data provided in our work from the colorimetric in vitro assay (standard in vitro cell viability assays according to ISO 10993-5). This can be considered in future works for a final validation of a formulation acting as a carrier for our promising nanosystems and providing more detailed characterization. In the present work, the main focus was to validate the technique in the production of the stable Zein-honey-chitosan nanoparticles, which can be used for different biomedical applications. Depending on the intended application for the nanoparticles, more specific and detailed assays must be hypothesized. Considering the suggestions of the reviewer, this has been included in future perspectives in the revised version of the manuscript (please see page 12, lines 436-437).
References
Arenas-Guerrero, P., Delgado, Á. V., Donovan, K. J., Scott, K., Bellini, T., Mantegazza, F., & Jiménez, M. L. (2018). Determination of the size distribution of non-spherical nanoparticles by electric birefringence-based methods. Scientific reports, 8(1), 1-10.
Liu, T., & Xiao, Z. (2012). Dynamic light scattering of rigid rods–a universal relationship on the apparent diffusion coefficient as revealed by numerical studies and its use for rod length determination. Macromolecular Chemistry and Physics, 213(16), 1697-1705.
Luo, Y., Zhang, B., Cheng, W.-H., & Wang, Q. (2010). Preparation, characterization and evaluation of selenite-loaded chitosan/TPP nanoparticles with or without zein coating. Carbohydrate Polymers, 82(3), 942-951.
Luo, Y., Zhang, B., Whent, M., Yu, L. L., & Wang, Q. (2011). Preparation and characterization of zein/chitosan complex for encapsulation of α-tocopherol, and its in vitro controlled release study. Colloids and Surfaces B: Biointerfaces, 85(2), 145-152.